# communications
# engineering

# Liquid-shaped microlens for scalable production of ultrahigh-resolution optical coherence tomography microendoscope

Chao Xu [1], Xin Guan[2], Syeda Aimen Abbasi [1], Neng Xia[3], To Ngai[2], Li Zhang [3], Ho-Pui Ho[1], Sze Hang Calvin Ng[4] & Wu Yuan [1✉]

Endoscopic optical coherence tomography (OCT) is a valuable tool for providing diagnostic images of internal organs and guiding interventions in real time. Miniaturized OCT endoscopes are essential for imaging small and convoluted luminal organs while minimizing invasiveness. However, current methods for fabricating miniature fiber probes have limited ability to correct optical aberrations, leading to suboptimal imaging performance. Here we introduce a liquid shaping technique for the rapid and scalable fabrication of ultrathin and high-performance OCT microendoscopes suitable for minimally invasive clinical applications. This technique enables the flexible customization of freeform microlenses with sub-nanometer optical surface roughness by regulating the minimum energy state of curable optical liquid on a wettability-modified substrate and precisely controlling the liquid volume and physical boundary on a substrate. Using this technique, we simultaneously fabricated 800-nm OCT microendoscopes with a diameter of approximately 0.6 mm and evaluated their ultrahigh-resolution imaging performance in the esophagus of rats and the aorta and brain of mice.

[1] Department of Biomedical Engineering, The Chinese University of Hong Kong, Hong Kong SAR, China. [2] Department of Chemistry, The Chinese University of Hong Kong, Hong Kong SAR, China. [3] Department of Mechanical and Automation Engineering, The Chinese University of Hong Kong, Hong Kong SAR, China. [4] Department of Surgery, The Chinese University of Hong Kong, Hong Kong SAR, China. ✉email: wyuan@cuhk.edu.hk

Endoscopic optical coherence tomography (OCT) is a label-free technology that enables the real-time, three-dimensional (3D) in vivo imaging of luminal organs with an imaging depth of 1–3 mm in tissues[1,2]. This technique allows the near-histologic quality visualization of tissue microstructures and provides histopathological information. Endoscopic OCT overcomes the limitations of traditional biopsy by enabling volumetric sampling across a large area without necessitating tissue removal[3]. Most current endoscopic OCT systems typically operate at 1300 nm and have a limited resolution of approximately 10 μm[3–7]. To address these limitations, endoscopic OCT operating at 800 nm has been developed. This OCT offers a considerably high resolution (~2–4 μm) and enhances image contrast, albeit at the expense of a shallower imaging depth compared to the 1300-nm OCT[8–11]. This technology has been demonstrated to be potentially valuable for evaluating the airways[12,13], gastrointestinal tract[9,11,14], blood vessels[15], and cervixes[16–18], et al.

When imaging small and convoluted luminal organs, such as narrow arteries and peripheral bronchioles, a miniaturized OCT endoscope that offers ease of access and adequate mechanical flexibility without causing discomfort or damage to tissues is required[9,13,19]. A small probe can extend the limited imaging depth of OCT, facilitating minimally invasive interstitial imaging in solid tissues and organs, such as the brain[20]. Therefore, an 800-nm OCT endoscope with an ultrahigh resolution (e.g., an axial resolution of <3 μm in the air) and ultrathin form factor (e.g., an outer diameter of <1 mm) is required for accurately detecting subtle pathological changes in tissues while ensuring minimal invasiveness during imaging.

To date, most miniature OCT probes have been fabricated using all-fiber distal focusing optics, typically composed of a delivery fiber spliced with a graded-index (GRIN) fiber and an angle-polished, side-deflecting fiber mirror[21] or a fiber ball lens directly melted and angle-polished on the fiber tip[9]. However, a conventional OCT microendoscope composed of GRIN fiber suffers from severe chromatic aberration at 800 nm. Previous achromatic design using a diffractive lens suffers from relatively low transmission efficiency[22,23]. Although an achromatic 800-nm OCT microprobe can be developed using an angle-polished distal fiber ball lens, the fiber-melting technique does not provide adequate flexibility to fabricate a ball lens with optimal balance among working distance, resolution, and depth of focus (DOF)[9,20]. Additionally, the fabrication of a beam reflector in a fiber ball-lens-based OCT probe involves the use of an angle-polishing procedure that is labor-intensive and presents challenges in achieving optimal optical surface roughness. Recently, two-photon 3D microprinting has been employed to construct freeform side-deflecting micro-optics on a single-mode fiber, resulting in a 1300-nm OCT endoscope with a diameter of approximately 0.5 mm[19]. However, this microprinting technique is expensive and lacks potential for scalability. Moreover, the surface roughness of 3D-printed optics is approximately 10–200 nm, which is not ideal for OCT imaging[19,24].

In this study, we introduce a liquid shaping technique that facilitates the rapid simultaneous fabrication of side-focusing microlenses on fiber tips for use in ultrahigh-resolution OCT microendoscopy. By regulating the minimum energy state of curable optical liquid on a substrate surface with tailored wetting properties, as well as controlling the droplet's volume and its physical boundary on the substrate, the size and shape of the distal microlens of an OCT probe can be flexibly customized using our method. This helps correct chromatic aberration and optimize imaging performance. This technique also eliminates the need for angle-polishing and results in liquid-shaped microlens with sub-nanometer surface roughness. Using this technique, we successfully fabricated ultrathin, aberration-corrected 800-nm OCT microendoscopes simultaneously. The resulting endoscopes had a diameter of approximately 0.6 mm (including a protective sheath) and provided an ultrahigh resolution of 2.4 μm × 4.5 μm (in axial and transverse directions in air). Furthermore, we demonstrated the imaging performance, mechanical flexibility, and minimal invasiveness of these microendoscopes by imaging the esophagus of rats and the aorta and brain of mice. Our method potentially facilitates the scalable fabrication of cost-effective, high-performance OCT microendoscopes for minimally invasive and ultrahigh-resolution optical biopsies in clinical settings.

## Results

**Liquid shaping technique.** Freeform microlenses can be fabricated by manipulating the minimum energy state of curable optical liquid droplets on a wettability-modified substrate[25,26] (Fig. 1a(i)) and used as focusing micro-optics in OCT endoscope of ultrasmall size, i.e., OCT microendoscope, for volumetric optical biopsy in complex luminal organs (see Fig. 1a(ii) and Materials and methods). In our work, a piezoelectrically actuated dispenser was used to precisely control the volume of the optical liquid droplet (NOA 81, Norland Product Inc.). Thermal control was employed by heating the dispenser nozzle to approximately 75 °C to reduce the liquid viscosity from 300 to 30 cps, facilitating easy dispensing of the optical liquid[27,28] (see Fig. 1a(i) and Materials and methods). This approach enables the production of lenses of various shapes and sizes within tens of minutes by controlling a substrate's wettability (or surface energy, see Fig. 1b and Materials and methods) as well as the liquid volume (Fig. 1c) and physical boundaries (i.e., circular or elliptical, Fig. 1d, e) on a substrate. Furthermore, the liquid-shaped microlens provides a reflective surface with sub-nanometer surface roughness[29,30] (Fig. 1a(i)), eliminating the additional angle-polishing process required in conventional methods based on GRIN fiber and fiber ball lens.

The versatility of this technique was validated through the continuous adjustment of the contact angle of approximately 50 nL of optical liquid on a glass substrate (which has no physical boundary for the liquid) from about 50° to 110° at an angle precision of about 0.5°. This was achieved by altering the substrate's wettability using a previously reported fluorination-based surface wettability modification method[31,32] (see Fig. 2a and Materials and methods). Once the desired contact angle (such as 90°) of the microlens was confirmed, the lens radius ranging from 120 to 720 μm was achieved at a resolution of approximately 1 μm by precisely controlling the droplet volume with a precision of approximately 0.1 nL (see Fig. 2b and Materials and methods), thereby verifying the scale-invariant nature of the liquid shaping technique. The measured radii (and heights) of microlenses made of different liquid volumes on glass substrates with defined wettability and contact angles closely aligned with the theoretically calculated radii[33] (Fig. 2c). This indicates the satisfactory controllability of the liquid shaping technique in creating spherical and aspherical microlenses of predefined shapes and sizes on substrates without physical boundaries.

A physical boundary on a substrate can be utilized to fabricate lenses with other 3D shapes (Fig. 1d, e). For instance, a spheroid lens can be fabricated using a 3D-printed circular cylinder substrate (see Materials and methods and Movie S1). The radius of the liquid lens and contact angle are initially governed by the substrate's wetting property and liquid volume (see the blue, purple, and red solid lines in Fig. 2d) and subsequently determined by the physical boundary size in conjunction with

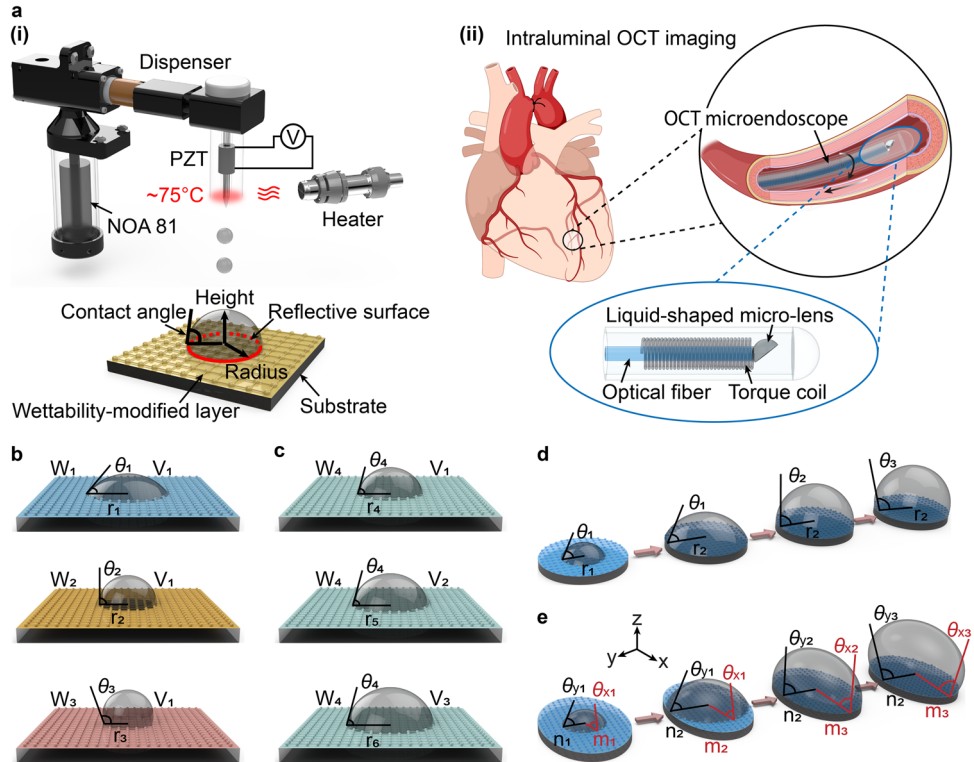

**Fig. 1 Schematic illustration of optical microlens fabrication using liquid shaping technique. a**(i) Production of microlens on a wettability-modified substrate using a piezoelectrically actuated dispenser with thermal control. **a**(ii) Ultrasmall OCT microendoscope based on liquid-shaped microlens for in vivo imaging in small and convoluted luminal organs, such as coronary vasculature. **b, c** Schematic of optical liquid on a substrate (without physical boundary). The microlenses of the same liquid volume ($V_1$) on the substrates of different wettability ($W_1$ - $W_3$) have a different radius, i.e., $r_1$ - $r_3$, and contact angle, i.e., $\theta_1$ - $\theta_3$ (**b**). As the liquid volume increases from $V_1$ to $V_3$ on the substrate of the same wettability $W_4$, the radius of microlens increases from $r_4$ to $r_6$ while maintaining a contact angle of $\theta_4$ (**c**). **d** Production of spheroid microlens using a substrate with a circular boundary. The radii and contact angles of microlenses made with different liquid volumes are indicated with $r_1$, $r_2$, and $\theta_1$, $\theta_2$, $\theta_3$, respectively. **e** Fabrication of ellipsoid microlens on a substrate with an elliptical boundary. The semi-major length, i.e., $m_1$, $m_2$, $m_3$, semi-minor length, i.e., $n_1$ and $n_2$, contact angle on x-z plane from $\theta_{x1}$ to $\theta_{x3}$, and contact angle on y-z plane from $\theta_{y1}$ to $\theta_{y3}$ of the microlenses are illustrated when different liquid volumes are used on the substrate. Please note that $n_1 = m_1$, $n_2 = m_2$, $\theta_{x1} = \theta_{y1}$, and $\theta_{x2} = \theta_{y2}$.

liquid volume (see the orange, green, and black solid lines in Fig. 2d).

The fabrication of an ellipsoid lens was validated using a 3D-printed elliptical cylinder substrate (see Materials and methods and Movie S2). With the increase in liquid volume, the shape and size of the lens are initially controlled by the substrate's wetting property and liquid volume (see the blue, purple, and red solid lines on x-z plane and blue and purple dashed lines on y-z plane in Fig. 2e) and then primarily by the physical boundary constraint (see the orange, green, and black solid lines on x-z plane and the red, orange, green, and black dashed lines on y-z plane in Fig. 2e).

The polymerization of microlenses was performed in approximately 30 min using an ultraviolet (UV) lamp operating at a wavelength of around 365 nm and a power density of 65 mW/cm$^2$ (on a lens sample). A constant shrinkage ratio of 7.69% in volume was identified for the fully polymerized microlenses, which kept almost unchanged contact angle and lens profile as those before polymerization[34,35] (see Fig. 2f and Materials and methods). Therefore, the calculated volume of the designed microlens was increased by 8.33% to compensate for the shrinkage effect and thus achieve the desired shape and size of the polymerized microlens (see Supplementary Note 1 and Supplementary Table 1).

**Liquid-shaped OCT microendoscope.** The liquid shaping technique facilitates the fabrication of an ultrathin OCT fiber probe,

which consists of a single-mode fiber, a non-core fiber (NCF), and a custom liquid-shaped microlens (Fig. 3a(i)). Initially, an NCF piece is spliced to a single-mode fiber and then coupled to a polymerized microlens at an incident angle $\theta$ by using NOA 81 as an optical adhesive (see Materials and methods). Subsequently, the OCT probe is protected with a hypodermic tube (rigid version) or a torque coil (flexible version). The entire probe is then encased in a protective sheath, such as a glass capillary tube (for the rigid version) or a plastic sheath (for the flexible version), to form an OCT microendoscope (Fig. 3a(ii)). This technique enabled the scalable production and fabrication of five microendoscopes simultaneously (see Materials and methods and Fig. S1).

To demonstrate the feasibility of the liquid shaping technique for imaging performance optimization and aberration correction, we designed an 800-nm OCT microendoscope using a simple semi-spherical microlens (Fig. 3a(iii)). The light incident angle was initially optimized to minimize light back-reflection in the microendoscope by performing stray light analysis in OpticStudio (see Materials and methods). We used the total internal reflection with an incident angle above the critical angle (i.e., approximately 40.2° at 842 nm) on the reflective surface of the microlens. The simulation results indicated that back-reflection in the OCT probe was mainly determined by the incident angle and lens radius, whereas its dependence on the NCF length was relatively low (Fig. 3b). Generally, a larger lens radius and incident angle result in lower back-reflection in the OCT probe. Because a small microlens is preferred to fabricate a miniature probe and a larger

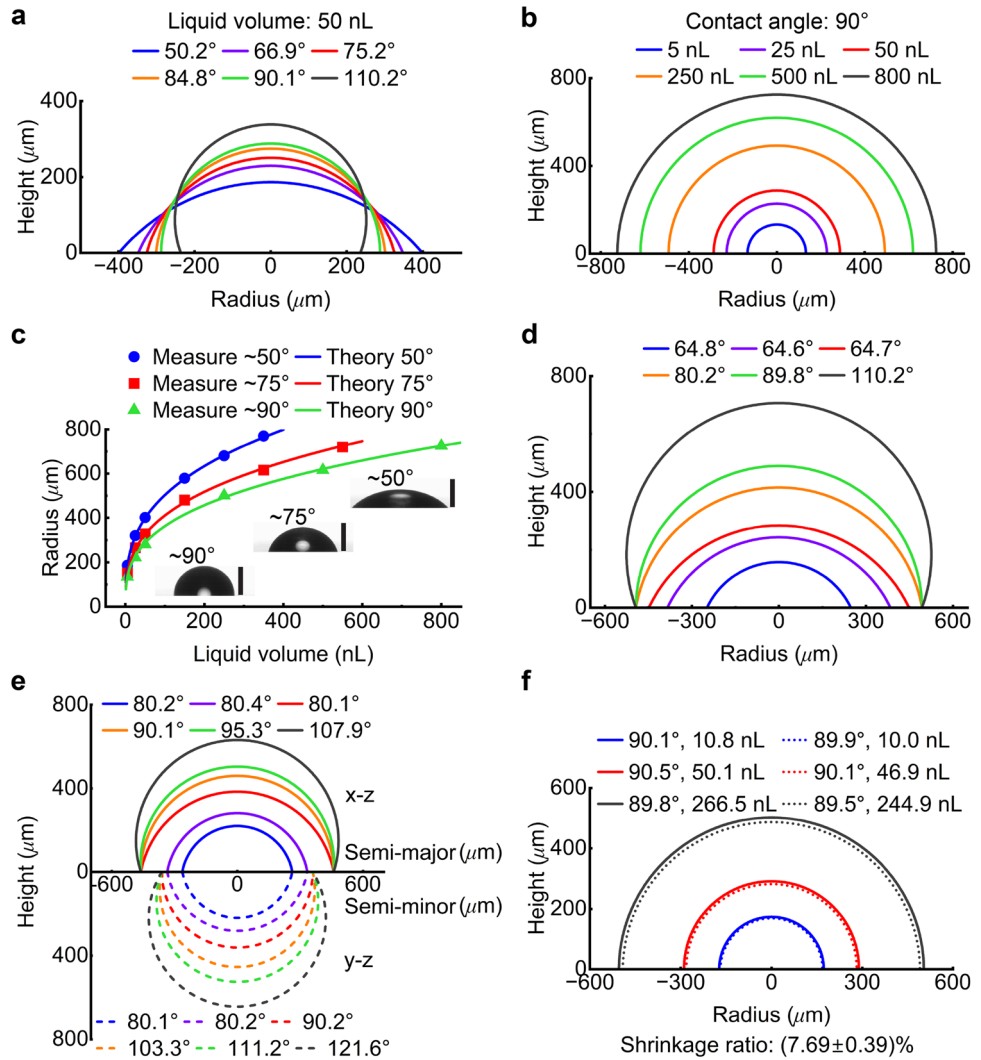

**Fig. 2 Shape and size control of the microlens using liquid shaping technique. a** The contact angles and sizes (in radius and height) of microlenses (illustrated in cross-sections) regulated by the wettability of glass substrate (without physical boundary) while using a constant liquid volume of about 50 nL. **b** The microlenses of constant 90° contact angle (in cross-sectional view) with their sizes (in radius and height) controlled by the droplet volumes from 5 nL to 800 nL. **c** Comparison of measured and theoretically calculated radii of microlenses regulated by the liquid volume on the substrates of different wettability (indicated with colors). The green triangle represents the same set of measurements shown in (**b**). The insets show three representative microlenses of contact angles and radii of about 50°/440 μm, 75°/334 μm, and 90°/277 μm, respectively. Scale bars are 250 μm. **d** Cross-sectional profiles of spheroid microlenses made on a substrate of a circular boundary (i.e., a radius of 493 μm). **e** Cross-sectional profiles of ellipsoid microlenses fabricated on a substrate of an elliptical boundary (i.e., a semi-major length of 460 μm and a semi-minor length of 360 μm), the shapes and sizes of microlenses are displayed on both x-z (colored solid lines) and y-z planes (colored dashed lines). **f** Study of the shrinkage effect in microlenses (in terms of contact angle and volume) by comparing their cross-sectional profiles before (colored solid lines) and after (colored dotted lines) ultraviolet light-induced polymerization. The shrinkage ratio is calculated using $(V_1-V_2)/V_1$, where $V_1$ and $V_2$ are the lens volume before and after polymerization.

incident angle leads to a smaller numerical aperture and larger focused spot size at a fixed working distance, we selected 52.5° in our design to achieve a back-reflection of less than −56 dB for a lens radius from 125 to 150 μm (Fig. 3c).

To further optimize imaging performance, we calculated the chromatic focus shift (within a spectrum ranging from 750 to 950 nm), focused spot size (the average spot size measured in x and y directions on the focal plane), astigmatism ratio (the ratio of spot sizes measured in x and y directions on the focal plane), effective DOF[19], and working distance for different combinations of NCF length and lens radius using OpticStudio (Fig. 3d–h and Materials and methods). It is noted that the source of astigmatism is derived from the cylindrical wall of the glass capillary or plastic sheath, which has different curvatures parallel and perpendicular to the endoscope axis[19,36–38]. Compiling the

simulated results indicated an optimal design region (Fig. 3i) that provides optimal imaging performance, such as a chromatic focal shift of less than 6 μm, a focused spot size (i.e., transverse resolution) of less than 6 μm, an astigmatism ratio between 0.9 and 1.1, an effective DOF of larger than 150 μm, and a working distance of between 200 and 300 μm. We used a design that involves fabricating a liquid-shaped microendoscope using a 500-μm-long NCF, a microlens with a 140-μm radius, and an incident angle of 52.5°. This approach enables achromatic performance to be achieved with a minimal focal shift of approximately 5.4 μm, a high transverse resolution of approximately 4.6 μm, a low astigmatism ratio of approximately 1.05 on the focal plane, a low back-reflection of less than −56 dB, and an appropriate DOF and working distance of approximately 197.3 and 238 μm, respectively (Fig. 3i).

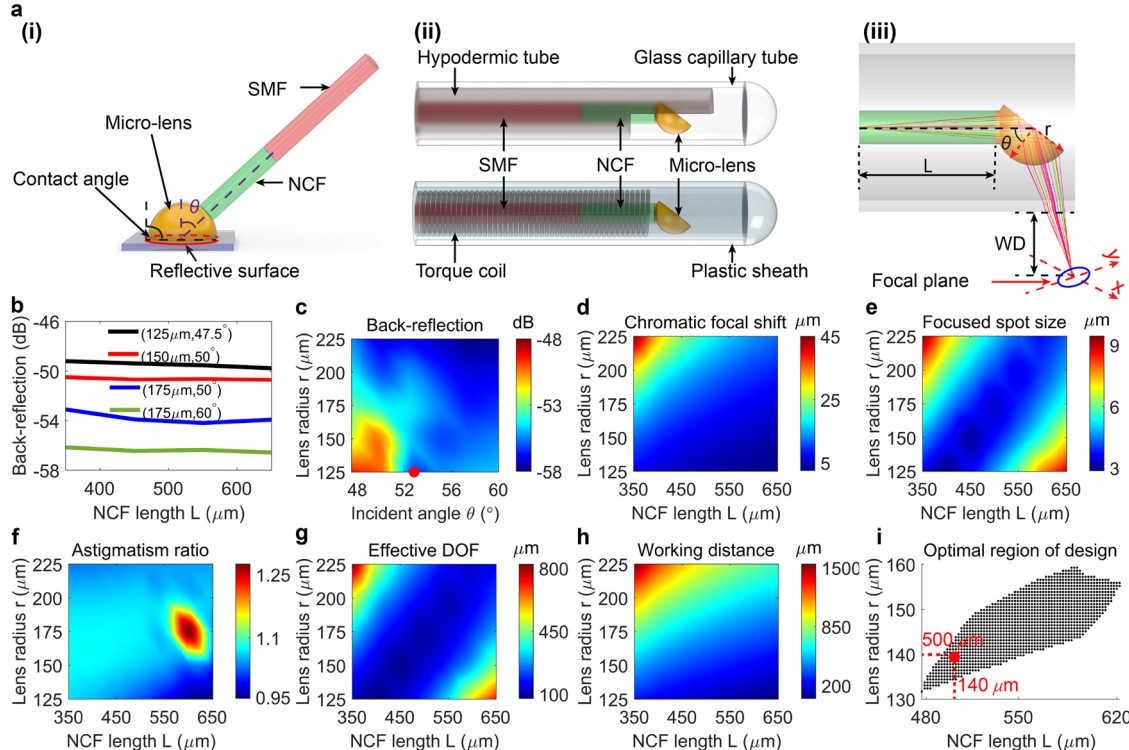

**Fig. 3 Liquid-shaped microendoscope. a**(i) Fabrication of liquid-shaped OCT fiber probe. Optical liquid of pre-calculated volume on a glass substrate with a modified wetting property forms a microlens of custom contact angle, shape, and size, which is then polymerized and used as distal optics of fiber probe made of single-mode fiber (SMF) and non-core fiber (NCF). Here an incident angle $\theta$ is formed in between the OCT light axis with respect to the normal axis of the reflective surface of microlens. **a**(ii) Schematic of OCT microendoscope. The fiber probe is guarded in a hypodermic tube (rigid version) or a torque coil (flexible version) and protected with a glass capillary tube (rigid version) or a plastic sheath (flexible version). **a**(iii) Zoomed-in view of the distal part of the microendoscope, consisting of an NCF of length L and a semi-spherical microlens of radius r and utilizing an incident angle $\theta$. Here the working distance (WD) is measured as the normal distance between the protective sheath surface and the center of the focal plane. **b** Calculated back-reflection in microendoscope versus NCF length L for four representative combinations of lens radius r and incident angle $\theta$. The back-reflection is found to decrease by about 0.55 dB (black line), 0.33 dB (red line), 0.89 dB (blue line), and 0.41 dB (green line), respectively, as NCF length L increases from 350 μm to 650 μm for each combination. **c** Calculated back-reflection versus lens radius r and incident angle $\theta$ in the microendoscope using an NCF length of 500 μm. The desired incident angle of 52.5° was indicated by a red dot. Chromatic focal shift (**d**), focused spot size (**e**), astigmatism ratio (**f**), effective depth of focus (DOF, **g**), and working distance (**h**) calculated at different NCF lengths L and lens radii r when the incident angle is 52.5°. **i** Optimal design region compiled from the calculated results in (**d**–**h**). The design adopted to fabricate the microendoscope is indicated with a red dot.

**Characterization of the OCT microendoscope.** Both the rigid and flexible versions of liquid-shaped 800-nm OCT microendoscopes, with diameters of approximately 0.6 mm (including a protective sheath), were fabricated using the aforementioned probe design (see Fig. 4a, b and Materials and methods). Five microendoscopes were simultaneously fabricated in 90 min, with comparable results, demonstrating the scalability of the proposed method (see Materials and methods, Supplementary Note 2, Supplementary Table 2 and Figs. S1 and S2). To illustrate the fabrication quality and controllability of the liquid shaping technique, the surface profile of the distal semi-spherical microlens was first characterized using a 3D noncontact confocal surface profiler (MarSurf CM Expert, Mahr Inc.). Because the profiler's imageable slope angle was limited to only approximately 45°, we measured only the top 40-μm height of the lens (Fig. 4c). The fitted one-dimensional surface profiles at four azimuthal angles (i.e., 0°, 45°, 90°, and 135°) indicated a microlens of a symmetric semi-spherical shape with a radius of approximately 140.5 ± 0.5 μm (close to the designed radius of 140 μm), suggesting the precise control of lens shape and size (Fig. 4d). Using a white-light interferometry-based surface profiler (MarSurf WI 50, Mahr Inc.), we measured the surface roughness of the microlens on the top curved surface to be 0.84 ± 0.11 nm (Fig. 4e). This sub-nanometer roughness is due to the smoothness of the

liquid-air interface in the liquid shaping technique[29,30]. Meanwhile, we noted a surface roughness of 0.53 ± 0.11 nm on the flat reflective surface of the microlens, owing to the use of an ultra-flat glass substrate (see Fig. 4f and Materials and methods). A liquid-shaped microlens with sub-nanometer surface roughness enables the effective mitigation of unwanted light scattering in the distal focusing optics of the OCT microendoscope.

To further characterize microendoscopes, we constructed an 800-nm spectral-domain OCT (SD-OCT) system. The configuration of this constructed system was similar to those developed previously[8,9,20] (see Supplementary Note 3 and Fig. S3). The diameter of the OCT laser beam exiting the protective sheath, such as the glass capillary tube used in the rigid microendoscope, was measured in both x and y directions along the light-emitting direction using an optical beam profiler (BladeCam2-XHR, DataRay Inc., Fig. 4g). The smallest focused spot was measured approximately 240 μm away from the outer sheath surface, exhibiting spot sizes of about 4.5 and 4.3 μm in x and y directions, respectively (with a mean diameter of 4.5 μm, inset of Fig. 4g), indicating a low astigmatism ratio of 1.05 on the focal plane. These measurements align well with the simulated focused spot size of 4.6 μm at a working distance of 238 μm. Furthermore, the measured effective DOF was approximately 200 μm, which was estimated by polynomial-fitting the mean beam diameters within

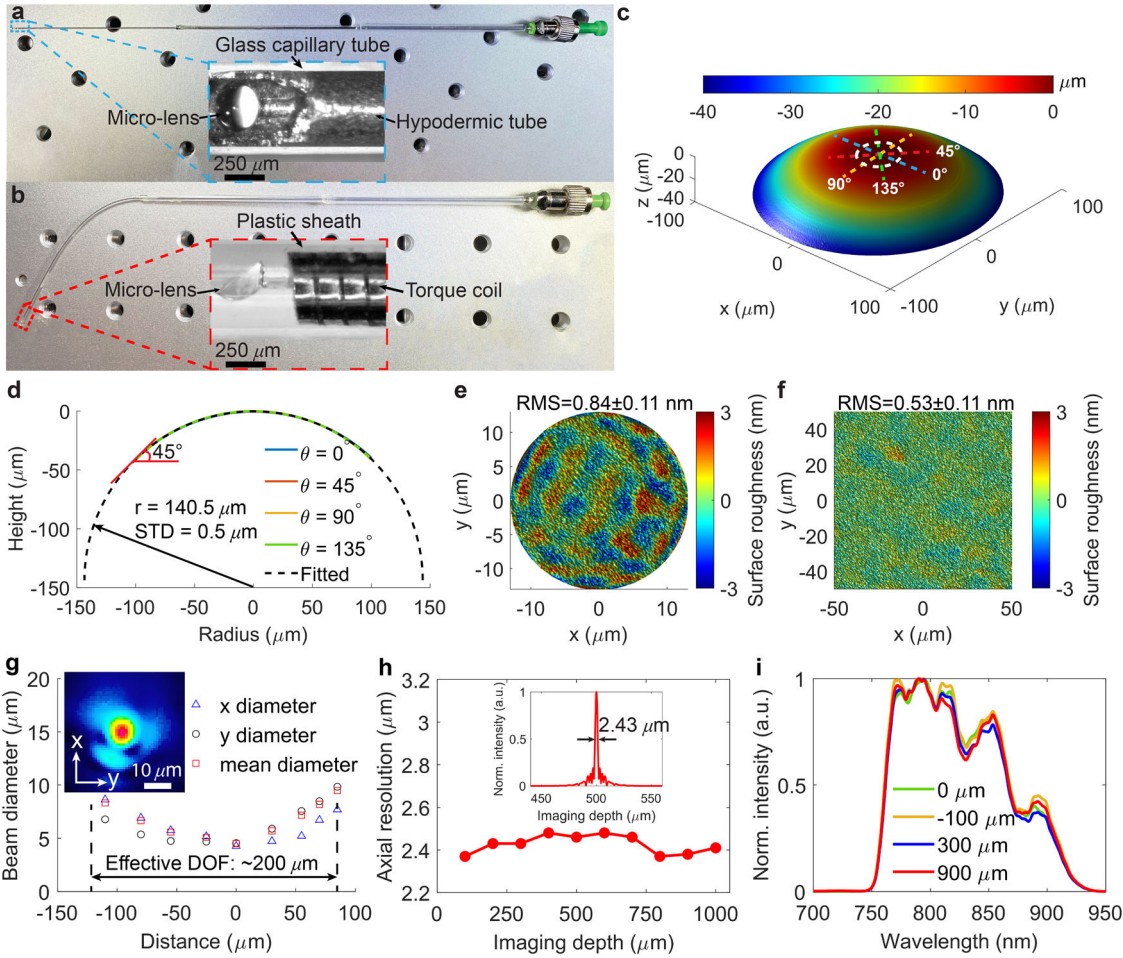

**Fig. 4 OCT microendoscope characterization. a** Photograph of a rigid microendoscope and zoomed-in view of its distal optics (inset). **b** Photograph of a flexible microendoscope and zoomed-in view of its distal optics (inset). **c** Measured 3D profile of the liquid-shaped microlens used in microendoscope. **d** 1D surface profiles of microlens extended by nonlinear least square fitting the measurements along four representative azimuthal angles at 0°, 45°, 90°, and 135° (blue, red, orange, and green dashed lines in **c**). **e** Representative surface roughness measured on the top curved surface of the microlens (white dashed circle in **c**). **f** Representative surface roughness measured on the flat reflective surface of microlens. **g** Representative focused laser beam profile and focused laser spot (inset) measured out of a 0.6-mm protective glass capillary tube of the rigid microendoscope. The x-axis represents the distance relative to the focal plane along the light-emitting direction, and 0 μm is the location of focal plane. The beam diameters in x and y directions are measured using an optical beam profiler, and the mean diameters are calculated using weighted equations provided by DataRay, i.e., w = 0.83114 × x + 0.16886 × y when x ≥ y, or w = 0.16886 × x + 0.83114 × y when x < y; w is the mean diameter, and x and y indicate the beam diameters in x and y directions, respectively. **h** Axial resolution measured along the imaging depth with a representative point spread function (inset). Here 0 μm in imaging depth is where the zero optical-delay position between the reference and sample arms is located. **i** Back-reflected spectra obtained by moving a mirror along the light-emitting direction and measured out of the protective sheath of microendoscope. Here 0 μm indicates the position of focal plane.

the depth range where the beam diameter was smaller than twice the size of the focused spot[19]. Additionally, the achromaticity of the microendoscope was verified based on the observation of less than 5% variation in the axial resolution of approximately 2.43 μm along the 1-mm imaging depth (Fig. 4h) and confirmed by the nearly unchanged back-reflected spectra measured by moving a mirror along the light-emitting direction[9] (Fig. 4i). These results demonstrated the ultrahigh resolution of the liquid-shaped OCT microendoscope operating at 800 nm.

**Ultrahigh-resolution imaging of small luminal organs**. Our flexible microendoscope is advantageous in terms of its ultrathin size, mechanical flexibility, and ultrahigh resolution. It can pass through the constricted lumen, such as narrowed sections in the blood vessels, and smoothly scan small luminal organs, such as an infant's esophagus. Our microendoscope enables the detection of fine microstructures and subtle pathologies of diseased tissues in vivo. In this study, a rat's esophagus was imaged to evaluate the

imaging performance of the microendoscope. The probe initially traveled through the oral cavity of the rat, passed the pharynx, traversed the tight upper esophageal sphincter (a narrow luminal structure that facilitates swallowing and reduces food backflow into the pharynx), and finally reached the small esophagus. 3D volumetric imaging was performed at a speed of 10 frames/second over a 36-mm-long esophagus. A separation of adjacent frames, i.e., pitch number, of 20 μm was used to control the pullback speed of the microendoscope.

The reconstructed 3D volumetric image and the representative OCT cross-section clearly revealed the layered tissue structures of rat's esophagus (Fig. 5a, b). The fine laminar structures of the esophagus, including the stratified squamous epithelium (EP), lamina propria (LP), muscularis mucosae (MM), submucosa (SM), circular muscle (CM), and longitudinal muscle (LM), were clearly observed in a zoomed-in view (Fig. 5c). The microstructures of the esophagus observed on the OCT image aligned with the corresponding hematoxylin and eosin (H&E) histology

**Fig. 5 Imaging rat esophagus using flexible microendoscope. a** Cut-way view of a reconstructed 3D OCT image of a 36-mm-long rat esophagus.
**b** Representative 2D OCT image corresponding to the cross-section boxed with green dashed lines in (**a**). **c** 3x close-up view of the region labeled with red
dashed box in (**b**). **d** Correlated hematoxylin and eosin (H&E) histology. EP stratified squamous epithelium, LP lamina propria, MM muscularis mucosae,
SM submucosa, CM circular muscle, LM longitudinal muscle. Scale bars are 250 μm.

micrograph (Fig. 5d). Compared with conventional 1300-nm
OCT endoscopes, our liquid-shaped OCT microendoscope
operating at 800 nm enables the acquisition of ultrahigh-
resolution images of the fine microstructures in the esophagus.
This capability holds the potential for detecting subtle pathologies
associated with early-stage diseases in vivo[39].

**In situ imaging of narrow lumens in complex internal organs**.
Currently, the accurate assessment of high-risk arterial diseases,
such as atherosclerosis[40,41], in small vessels remains challenging
due to their narrow lumens, highly complex networks, and vast
distribution[42]. Thus, a flexible microendoscope that allows for
minimally invasive and ultrahigh-resolution imaging in small
blood vessels is highly desirable. To evaluate the functionality of
our microendoscope in the narrow lumens of blood vessels, we
performed the in situ imaging of the descending aorta, which has
a lumen diameter of approximately 0.6 mm, in a normal
mouse model.

In this study, the mouse was first euthanized by administering
an overdose of ketamine and xylazine before imaging. The mouse
heart and connected vessels were perfused with phosphate-
buffered saline using a 27-gauge needle inserted into the apex of
left ventricle[19,43]. This perfusion procedure depleted blood in the
aorta, thereby preventing any interference from high-scattering
red blood cells during OCT imaging. Subsequently, the micro-
endoscope was inserted through the left femoral artery. It passed
through the descending abdominal aorta and thoracic aorta
sections, finally reaching the section close to the aortic arch
(Fig. 6a).

A 14.6-mm-long section of the descending thoracic aorta was
imaged at a speed of 10 frames/second with a frame pitch of
20 μm (Fig. 6b). The imaged aorta was then excised, fixed,
embedded, sectioned into 10-μm-thick slices, and stained with
H&E. As demonstrated in the OCT cross-section, the zoomed-in
view, and the corresponding histology micrograph, we clearly
observed the laminar microstructures of the mouse aorta,
including the tunica intima (TI), tunica media (TM), tunica
adventitia (TA), and adipose tissues (AT) (Fig. 6c–e). Further-
more, multiple elastic lamellae (EL, with high scattering)
intermixed with smooth muscle sheets were identified in the
tunica media layer.

The ultrahigh-resolution imaging facilitated by the microendo-
scope enabled the clear delineation and accurate quantification of
the microstructures of the aorta for the in situ evaluation of
arterial diseases. A preliminary study revealed that in a normal
mouse, the thicknesses of the aortic layers were $11.0 \pm 1.2$,
$69.2 \pm 3.4$, and $28.2 \pm 2.2$ μm for the tunica intima, media, and
adventitia, respectively (Fig. 6f, g). In particular, the tunica

intima, which is the thinnest and innermost layer of an artery or
vein, is composed of a single layer of endothelial cells; its
thickening and proliferation are considered an early indication of
atherosclerosis[44,45].

**Minimally invasive interstitial imaging in deep brain in vivo**.
The ultrathin rigid microendoscope can extend the limited ima-
ging depth of OCT and perform high-resolution, volumetric
interstitial imaging of solid organs with minimal invasiveness.
This facilitates the access and evaluation of deep-seated diseases,
such as deep brain tumors, ischemic stroke, and epilepsy[46–49],
while minimizing the risks of hemorrhage and other trauma
caused by probe insertion[20].

To evaluate the functionality of the rigid microendoscope, we
performed in vivo deep-brain imaging in a mouse brain. After
making an incision on the scalp, we drilled two small burr holes
on each of the two contralateral sides of the skull (Fig. 7a). The
microendoscope was then inserted through the burr holes into
the deep brain, following an insertion trajectory perpendicular to
the brain surface (Fig. 7b). A 5-mm-long (in z direction)
cylindrical deep brain volume was imaged in 50 s at a speed of 10
frames/second. Because of the ultrahigh resolution of the
microendoscope, we clearly observed mouse brain structures,
including the cerebral cortex, corpus callosum, caudate putamen,
and ventral striatum, on the 3D volumetric image (Fig. 7c). The
*en face* projection view, generated by summing the unfolded
cylindrical brain volume along the imaging depth, revealed the
distinct laminated brain structures of the mouse, which were also
verified in the histology micrograph, indicating a good correlation
between the OCT representation and histomorphology (Fig. 7d, e
and S4). Representative OCT cross-sections acquired at different
depths illustrate the detailed morphological features of each
mouse brain structure (Fig. 7f–i). In particular, the filament
bundle structures of striatopallidal fibers in the caudate putamen
were clearly observed on the OCT images[20] (Fig. 4c, d, h).

**Discussion**
In comparison to the 1300-nm OCT, the 800-nm OCT provides
improved imaging resolution and contrast, albeit with a reduced
imaging depth. As conventional GRIN-fiber-based miniature
OCT endoscopes experience severe chromatic aberration at
800 nm, we previously developed an achromatic microprobe with
a diameter of 1 mm by utilizing a monolithic fiber ball lens with
the fiber melting technique[9]. However, to further minimize the
probe size to below 1 mm, the current fabrication technique
utilizing the fiber melting process is suboptimal, yielding a small
fiber ball lens with a low resolution and short DOF[9]. In addition,
the miniature probe has a limited numerical aperture, which

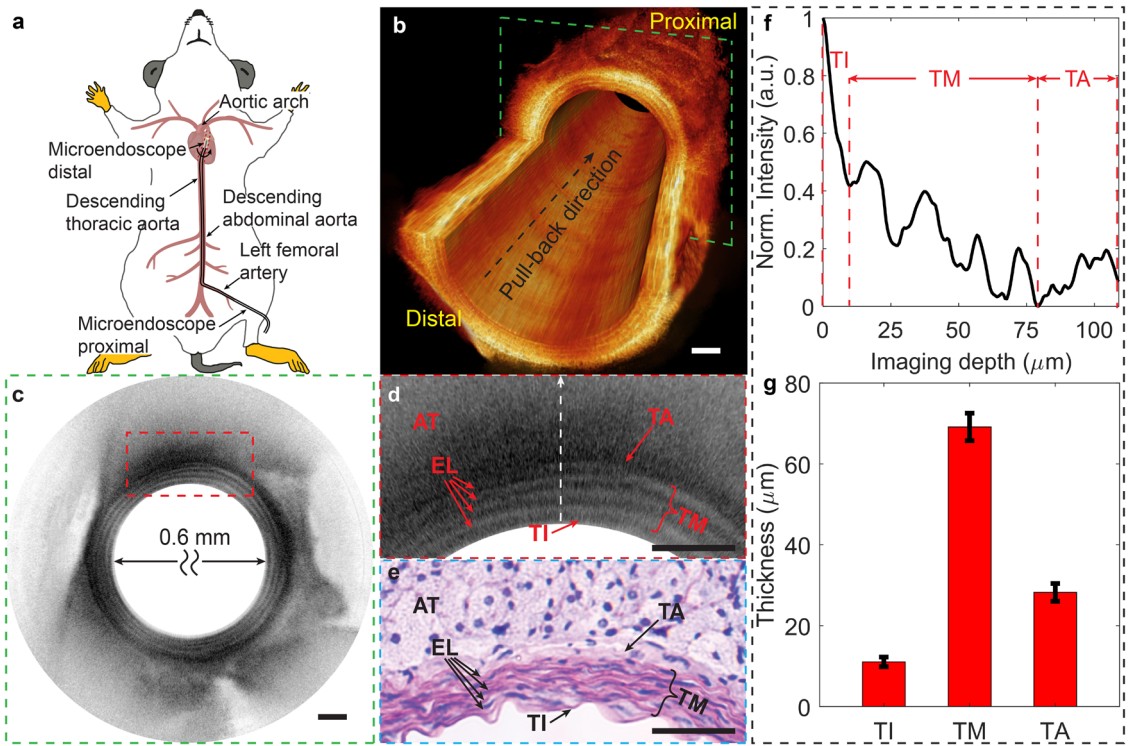

**Fig. 6 In situ imaging of mouse aorta with flexible microendoscope. a** A flexible microendoscope is deployed from the left femoral artery to the aortic arch in the mouse. **b** Cut-way view of a reconstructed 3D OCT image of a 14.6-mm-long mouse descending thoracic aorta. **c** Representative OCT cross section indicated with a green dashed box in (**b**). **d**, **e** 3x close-up view of the region boxed with red dashed lines in (**c**) and its correlated hematoxylin and eosin (H&E) histology. The white arrow points to the deep tissue. **f**, **g** Example quantification of each fine tissue layer's thickness (distance between red dashed lines) of mouse aorta along an A-line depth (the white arrow in **d**) (**f**). The thickness and its standard deviation of each aorta layer were measured using 40 A-lines from 10 representative OCT cross-sections (**g**). TI tunica intima, TM tunica media, TA tunica adventitia, EL elastic lamellae, AT adipose tissue. Scale bars are 100 μm.

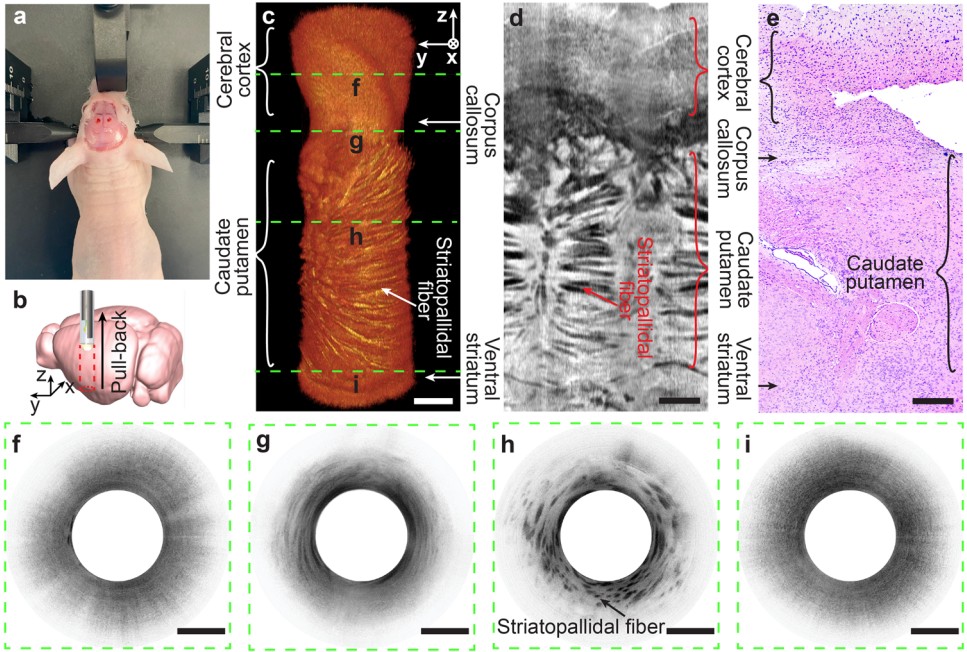

**Fig. 7 In vivo imaging of mouse deep brain with rigid microendoscope. a** The preparation of mouse brain for imaging with the scalp incised and two burr holes (about 1 mm in diameter) on skull. **b** Schematic of microendoscope imaging in mouse deep brain. **c** Reconstructed volumetric mouse brain image of 5-mm depth. **d** *En face* projection view of the unfolded cylindrical OCT volume shown in (**c**). **e** Hematoxylin and eosin (H&E) histology micrograph of brain sample correlated to (**c**, **d**). Representative cross-sectional images of different mouse brain structures, including cerebral cortex (**f**), corpus callosum (**g**), caudate putamen (**h**), and ventral striatum (**i**), corresponding to dashed lines in (**c**). Scale bars are 500 μm.

causes difficulty in achieving an optimal balance between high resolution and DOF.

An OCT microendoscope with a high resolution and large DOF is necessary for the accurate imaging of subtle pathological changes in tissues. For this purpose, an appropriately designed microlens is required in the microendoscope. However, it is difficult to precisely tailor a microlens by using the fiber melting technique. To overcome this limitation, we propose the liquid shaping technique. This method allows for the precise customization of the shape and size of a liquid droplet on a substrate, thereby producing a freeform microlens with an ultrasmall form factor, corrected aberrations, and desired imaging performance. These features can be achieved by conveniently modifying the wettability of a substrate, controlling the liquid volume, and utilizing physical constraints on the substrate. In contrast to the 3D printing method, our technique does not require expensive high-end machinery and can be easily scaled up. Additionally, the liquid shaping technique yields a custom microlens with a sub-nanometer surface roughness (Fig. 4), which is considerably better than the roughness (~10–200 nm) achieved using the two-photon 3D microprinting[19,24] (see Supplementary Table 3 for comparison).

The liquid shaping technique offers an approach to simultaneously fabricate high-performance OCT microendoscopes. The fabrication process of the microendoscope involves only standard and common optical fiber handling procedures, such as splicing and cleaving, and the simple gluing of a custom microlens to the fiber probe tip (see Materials and methods and Fig. S1). These processes can be streamlined for mass production. This approach eliminates the need for a time-consuming angle-polishing process that is required in conventional GRIN-fiber and fiber ball-lens-based methods. The use of our approach substantially reduces both fabrication time and cost and increases the yield rate due to the minimized reliance on human expertize (see Supplementary Table 3 for comparison). Furthermore, our longitudinal study revealed that the fabricated microendoscopes maintain their imaging performance over a long period (Fig. S5). Thus, the liquid-shaped microendoscope can serve as a low-cost and disposable OCT catheter for translational use.

In principle, the form factor of the microendoscope fabricated using the liquid shaping technique can be further minimized using a thinner hypodermic tube or torque coil and a smaller protective sheath over the entire probe. The imaging performance of a downsized microendoscope can be optimized by customizing a smaller microlens with appropriate back-reflection, achromaticity, astigmatism ratio, resolution, and DOF (Fig. 3). Likewise, the form factor of the microendoscope can be increased to image larger lumens (e.g., those with a diameter greater than 2 mm) by fabricating an appropriately designed microlens. In contrast, it would be suboptimal to fabricate a fiber ball-lens-based microprobe to image lumens larger than 2 mm because a microprobe with a longer working distance tends to lead to a degraded transverse resolution and, more importantly, an increased longitudinal focal shift[9].

Given that the conventional 1300-nm endoscopic OCT has successfully gained approval from the Food and Drug Administration for imaging the esophagus and coronary artery, the ultracompact and flexible liquid-shaped microendoscope operating at 800 nm presents promising opportunities for the in vivo ultrahigh-resolution imaging of tissue pathological changes in small and/or complex luminal organs, such as small airways and small blood vessels. The ability to obtain ultrahigh-resolution 3D microscopic images of a lumen enhances the likelihood of detecting premalignant or early-stage diseases. In addition, the decreased size and enhanced flexibility of the microendoscope enable sharp bending along the delivery path and substantially reduce the risk when used to image a small and delicate luminal organ. This ultrathin microendoscope can be potentially integrated within a biopsy needle, enabling the accurate assessment of the fine structures of target tissues. This would help guide biopsies, resulting in an improved diagnostic yield.

The liquid-shaped microendoscope is currently in its early stage of development, and substantial technical improvements can be achieved in the future. First, the current liquid shaping technique mainly relies on passive methods, such as substrate wettability, liquid volume, and physical boundary, to control the shape and size of the lens. Active methods that manipulate the thermal and/or electromagnetic field will be explored to gain new control dimensions for shaping microlens with complex freeform surfaces and advanced functionality[50,51]. Second, the current study demonstrated only the feasibility of fabricating five liquid-shaped microendoscopes simultaneously. We plan to further streamline and automate the fabrication process to increase the yield rate and reduce cost. In addition, we validated the imaging performance of the liquid-shaped microendoscope in small animals only. A systematic study of large animals is imperative to thoroughly validate the functionality and safety of our microendoscope before initiating clinical trials with patients. For future clinical use, the imaging speed of the microendoscope should be improved to minimize motion artifacts and achieve a short intervention time in the operating room. It is important to highlight that the proposed liquid shaping technique can be easily implemented in other types of fiberscopes that require a microlens on the fiber tip. This includes confocal, two-photon, and coherent Raman endoscopes, among others[52–54].

## Materials and methods

**Liquid droplet generation and characterization**. A piezo-electrically actuated dispenser (SA306, Sans Inc.) was utilized for droplet/microlens generation. This technique was reported in previous works and widely used for microarray printing[55], tissue engineering[28], and fabrication of functional materials[27]. To account for the system dispensing accuracy, the calibration of the dispensed liquid volume on the substrate was first performed (Fig. S6). A series of liquid volumes claimed by the dispenser was dispensed on the glass substrate to form the liquid microlens; the contact angle, dimensions, and volume of microlens were then measured using a liquid drop analysis system (OCA 25, Dataphysics Instruments GmbH) at room temperature of 25 °C and analyzed with SCA 20 software (DataPhysics Instruments GmbH). After calibration, the liquid microlens volume can be controlled at a precision of about 0.1 nL. The accuracy of contact angle measurement is ±0.1°, the accuracy of length measurement is ±0.5 μm, and the accuracy of volume measurement is ±0.05 nL.

As for the polymerized microlens, the contact angle, dimensions, and volume were characterized using the same method as liquid microlens, while the lens profile was measured using a confocal surface profiler (MarSurf CM Expert, Mahr Inc.) and its surface roughness was characterized with a white-light interferometry-based surface profiler (MarSurf WI 50, Mahr Inc.).

**Surface wettability modification of glass substrate**. Glass substrates (S2006A1, Ossila) used in our study provide a super-polished surface of about 1-nm roughness. To modify the surface wettability, the glass substrate is first treated in oxygen plasma bathing for 10 min. The substrate is then placed in a petri dish and soaked in the mixture of 1H, 1H, 2H, 2H–Perfluorooctyltriethoxysilane (POTS) (volume: 10–20 μL) and methylbenzene (volume: 10 mL). The petri dish is sealed and placed in a fume hood at room temperature for 4–16 h to achieve desired surface wettability and contact angle on the substrate (see

Supplementary Table 4). After that, the glass substrate is retrieved and thoroughly rinsed with absolute ethanol before use.

**The fabrication and surface wettability modification of cylinder substrates.** A stereolithography 3D printer (S300, nanoArch) was employed to fabricate cylinder substrates of circular or elliptical boundary with an about 400-μm height. Polyethylene Glycol Diacrylate (PEGDA) was used for 3D printing.

To modify the surface wettability, the cylinder substrate is first processed with oxygen plasma bathing for 10 min. Then, it is placed in a petri dish with the cylinder top surface immersed in 10 μL of POTS. The petri dish is sealed with Kapton tape and placed in a thermotank with a baking temperature of 100 °C for 4–8 h to achieve desired surface wettability and contact angle on the substrate (see Supplementary Table 5). After that, the cylinder substrate is retrieved and thoroughly rinsed with absolute ethanol before use.

**The fabrication of liquid-shaped microlens.** The critical procedures to fabricate microlens using liquid shaping technique are illustrated in Fig. S1a, b. Specifically, NOA 81 was selected over other optical liquids in this study (see Supplementary Note 4 and Supplementary Table 6 for rationale). The NOA 81 liquid was first degassed in a vacuum chamber (98 kPa vacuum level for 10 min) to remove any potential micro-bubbles. Then, the liquid of calculated volume was dispensed using the piezoelectrically actuated dispenser (SA306, Sans Inc.) on wettability-modified glass substrate or circular/elliptical cylinder substrates. The desired shape of size of microlens is obtained by precisely controlling liquid volume, wettability, and physical boundary of substrate.

After that, the liquid microlens on substrate was degassed again in a vacuum chamber (98 kPa vacuum level for 10 min). Finally, the liquid microlens was polymerized by using a UV lamp illumination for 30 min to ensure complete polymerization, which usually only needs 20 min. The UV lamp has a center wavelength of about 365 nm and provides power density of 65 mW/cm$^2$ on the lens sample. It should be noted that a polymerization-induced shrinkage effect of optical liquid (about 7.69%, see Fig. 2f, Supplementary Note 1, and Supplementary Table 1) was considered and pre-compensated in volume calculation during the microlens fabrication. For the microendoscopes demonstrated in current work, the lens liquid with a volume of about 6.2 nL was dispensed on the substrate to fabricate the semi-spherical microlens of about 140-μm radius and contact angle of about 89.8°.

**Zemax simulations.** OpticStudio (v17, Zemax LLC) was used to design and optimize the liquid-shaped microendoscope. The refractive index profiles and material dispersions of silica (of NCF) and NOA 81 are considered in simulations (Fig. S7). The back-reflection under different incident angles, lens radii, and NCF lengths was first calculated using stray light analysis in the non-sequential mode. After that, the ray-tracing simulation of the microendoscope was performed in the mixed sequential/non-sequential mode. By considering a single-mode fiber with numerical aperture of 0.13 and a spectrum ranging from 750 nm to 950 nm, the chromatic focal shift, focused spot size, astigmatism ratio, effective DOF, and working distance were calculated under different lens radii and NCF lengths.

**The simultaneous fabrication of microendoscopes.** The key procedures to fabricate microendoscopes are illustrated in Fig. S1c, d.

Essentially, a single-mode fiber (780HP, Thorlabs Inc.) was first spiced with a NCF (FG125LA, Thorlabs Inc.), which was then cleaved to a length of 500 ± 5 μm, to form a fiber probe. Both fiber splicing and cleaving procedures were performed using the same glass processor (GPX3800, Thorlabs Inc.) and five fiber probes were prepared in this step.

Second, the polymerized microlenses were coupled to fiber probes at an incident angle of 52.5°. Five microendoscopes were fabricated simultaneously with the homemade four-dimensional assembly setup (Fig. S1e). To ensure precise alignment between the fiber probe and microlens, we developed a high-precision four-dimensional assembly stage equipped with two inspection microscopes that offer top and side views. This stage initially establishes a predetermined incident angle between the fiber probe and microlens. Subsequently, the fine adjustment of the fiber-lens incident angle and distance is achieved with the assistance of the two microscopes. Following this, approximately 0.5 nL of NOA 81 is applied between the fiber tip and microlens to facilitate optical bonding, which is then cured using UV light. The fiber probe tip kept a safe distance of about 5 μm to microlens surface during the bonding procedure. 30-min UV light illumination (wavelength: 365 nm, power density on sample: 65 mW/cm$^2$) was used to achieve a complete polymerization of the bonding adhesive.

Third, five fiber probes with a bonded microlens were removed gently from the glass substrate with the aid of a razor blade. The resulting fiber probes usually afford a one-way transmission efficiency of at least 94%.

Finally, the 215-mm-long fiber probes were encased in 26-gauge hypodermic tubes of an open window at the end to make rigid microendoscopes. Silica glass capillary (Molex LLC) of about 617 μm in outer diameter and about 40 μm in wall thickness was employed to protect the microendoscope during imaging.

Flexible microendoscopes of 215 mm in length were fabricated using the similar procedures. After removing the fiber probes from the glass substrate, torque coils of 450 μm in outer diameter were then used to encase the imaging probes. The flexible microendoscope was protected with a fluorinated ethylene propylene (FEP) plastic sheath (Zeus Inc.) of a 626-μm outer diameter and a 45-μm wall thickness during imaging. The total fabrication time of the microendoscope was about 90 min.

**Animal studies and histological correlation.** The protocol of OCT endoscopic imaging in rat and mouse was approved by the Laboratory Animal Services Centre at The Chinese University of Hong Kong.

For rat esophagus ($n = 4$, Sprague Dawley rats) and mouse aorta ($n = 4$, nude mice) imaging, animals were euthanized by overdosing of ketamine (100 mg/kg) and xylazine (16 mg/kg) before OCT imaging. The flexible microendoscope was first deployed near to rat GEJ section and mouse aortic arch, respectively. Then, OCT pullback imaging was performed at a speed of 10 frames/second. The microendoscope probe was retreated after imaging, while the plastic sheath was left in the lumens for registration of the imaged tissues. The imaged esophagus and aorta were harvested and fixed in formalin together with plastic sheath overnight before being submitted for histological processing. Standard H&E slides were obtained and correlated with OCT imaging results.

For deep brain imaging ($n = 4$, nude mice), the mouse anesthetization was first introduced by intraperitoneal injection of ketamine (100 mg/kg) and xylazine (16 mg/kg) and further maintained by inhaling 2% isoflurane with medical oxygen. Burr holes (about 1 mm in diameter) were made on two contralateral sides of mouse skull to allow the deployment of rigid

microendoscope (Fig. 7a). The burr hole location was selected to avoid major blood vessels. Before insertion, a thin needle (with a diameter of about 300 μm) was utilized to introduce a small hole through the pia mater. The microendoscope was then inserted through the burr hole into the mouse brain with a slow speed of about 10 μm/s. After imaging, the mouse was sacrificed and the brain was immediately harvested. A glass capillary tube was inserted back into the brain tissue along the same imaging trajectory, helping register the imaged tissue. The brain tissue together with glass capillary was then placed in formalin for 48 h. After fixation, the brain tissue was divided into two sections (Fig. S8) for further histological processing. Standard H&E slides were obtained and correlated with *en face* OCT images.

**Statistics and reproducibility**. In the study of the mouse aorta with sample size of 4, the thickness and its standard deviation of each aorta layer, including the tunica intima, media, and adventitia, were measured using 40 A-lines from 10 representative OCT cross-sections to ensure the data reliability. In addition, the study of rat esophagus and mouse brain, the sample sizes are 4.

Five microendoscopes were simultaneously fabricated, including two rigid ones and three flexible ones. They were characterized of similar results and tested for OCT imaging with comparable image qualities, showing the scalability of the liquid shaping technique for microendoscope fabrication (see Supplementary Note 2 and Fig. S2).

**Reporting summary**. Further information on research design is available in the Nature Portfolio Reporting Summary linked to this article.

## Data availability

All data needed to evaluate the conclusions in the paper are present in the paper and/or the Supplementary Information. Additional data related to this paper may be requested from the corresponding author.

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

## Acknowledgements

The authors are grateful to Mr. Paul Zhou for his help with the microlens characterizations. This work was supported by the Research Grants Council (RGC) of Hong Kong SAR (ECS24211020, GRF14203821, GRF14216222), the Innovation and Technology Fund (ITF) of Hong Kong SAR (ITS/240/21), and the Science, Technology and Innovation Commission (STIC) of Shenzhen Municipality (SGDX20220530111005039).

## Author contributions

W.Y.: Conceived the idea; C.X.: Ran the simulations; designed, fabricated, and characterized the microendoscopes; W.Y., C.X.: Designed the animal experiments; C.X., X.G.: Conducted the surface processing of substrates; C.X., S.A.A.: Performed the animal experiments and histology; C.X., N.X.: Conducted the 3D printing of cylinder substrates; W.Y.: Supervised the theoretical and experimental work; T.N., L.Z., H.-P.H., S.H.: Contributed to the paper revision; All the authors contributed to the paper writing.

## Competing interests

The authors declare no competing interests.
