## [Peer Review File · Communications Engineering]

Reviewers' comments:

Reviewer #1 (Remarks to the Author):

In this manuscript, the authors report the application of liquid-shaping method to the fabrication of sub-millimeter side-focusing ball lens that is well-known as essential to a miniature rigid or flexible OCT catheter. The authors explained their design logic and performance characterization, and presented representative experimental results. The logic of the manuscript fits the scope of Comms. Eng., and the overall flow is easy to follow.

However, to minimize misleading information and to help future readers better appreciate the manuscript, I suggest the authors address the following comments.

1. The initial and key word “self-assembled” of the title appeared only twice in the main text, which reflects on the lack of the evidence on why and how the microlens is self-assembled. The overall liquid-shaping-based fabrication procedure clearly necessitate extensive human intervention. Please reconsider the common meaning of “self-assembling”, and then either provide more supportive explanation, or rectify the terminology to avoid misleading future readers.
2. Since the authors compared their fabrication procedure to two-photon 3D microprinting, please provide more comparison information on pros and cons of these two methods, e.g., the cost of equipment (i.e., liquid-shaping v.s. two-photon microprinting), the cost of material (including the glass substrate with wetting property modified used in this manuscript), and complexity of freeform surface that could be achieved, so that future readers could better understand which technology to choose for their application.
3. The design rationale presented in Fig. 3 needs more clarification.
 - 3.1 The word “semi-spheroid” here is potentially misleading. A spheroid, also known as an ellipsoid of revolution or rotational ellipsoid, is a quadric surface obtained by rotating an ellipse about one of its principal axes; in other words, an ellipsoid with two equal semi-diameters. Also, the word spheroid originally meant "an approximately spherical body". Here, however, based on the description in line 166-167, what the authors seem to mean and used in their simulation is a semi-spherical lens “of radius r ...”. Please double check and clarify/revise accordingly.
 - 3.2 The source of astigmatism as presented in Fig. 3f is not clear. If the (total internal) reflection surface coincides with a great circle of the semi-spherical lens (i.e., the center of the spherical surface lies within the reflection surface), the entire optical train reduces to an on-axis geometry which should not suffer from astigmatism. If the reflection surface is offset from the great circle (i.e., from the centre of the sphere), which seems to be quite possible based on information presented in Fig. 1 and Fig. 2, this offset plays a more important role than NCF length and lens radius as presented in Fig. 3f, and therefore should be explicitly presented and Fig. 3f (and probably many other subfigures in Fig. 3) should be re-plotted with the main text revised accordingly also.
 - 3.3 Still about astigmatism, one important culprit is the equivalent cylindrical lens formed by the wall of the glass capillary tube or the plastic sheath. Whether such astigmatism source is accounted for is unclear, and little information is given regarding the refractive index or dimensions of the glass capillary (except an OD of 617 μm , line 523) or plastic sheath.

4. Line 220, the word “radial angle” could be hard to understand. Please consider paraphrasing the description (e.g., how about azimuthal angle?)

5. Does the ~2.4 μm axial resolution measured in air or in tissue? Please clarify.

6. Regarding chromatic aberration and astigmatism correction in side-viewing OCT probes, there have been previous efforts that should be cited; just to give a few examples:

A. Jiefeng Xi, Li Huo, Yicong Wu, Michael J. Cobb, Joo Ha Hwang, and Xingde Li, "High-resolution OCT balloon imaging catheter with astigmatism correction," *Opt. Lett.* 34, 1943-1945 (2009)

B. Jessica Mavadia-Shukla, Payam Fathi, Wenxuan Liang, Shaoguang Wu, Cynthia Sears, and Xingde Li, "High-speed, ultrahigh-resolution distal scanning OCT endoscopy at 800 nm for in vivo imaging of colon tumorigenesis on murine models," *Biomed. Opt. Express* 9, 3731-3739 (2018)

C. Li, K, Liang, W, Mavadia-Shukla, J, et al. Super-achromatic optical coherence tomography capsule for ultrahigh-resolution imaging of esophagus. *J. Biophotonics*. 2019; 12:e201800205.

7. It is worthy of noting that the liquid-shaped microlens fabrication protocol is applicable to fiberscopes based on other imaging modalities like confocal, two-photon, and coherent Raman, etc., that could also benefit from advanced fiber-tip engineering. I suggest the authors include brief discussion about this topic to maximize the impact of their work, and here are some references to consider:

A. Jae-Beom Kim, Jaehun Jeon, Kyungmin Hwang, Daniel Y. Kim, Ki-Hun Jeong, "Objective-lens-free confocal endomicroscope using Lissajous scanning lensed-fiber," *J. Optical Microsystems* 1(3) 034501 (28 May 2021) <https://doi.org/10.1117/1.JOM.1.3.034501>

B. W. Liang et al., "Throughput-Speed Product Augmentation for Scanning Fiber-Optic Two-Photon Endomicroscopy," in *IEEE Transactions on Medical Imaging*, vol. 39, no. 12, pp. 3779-3787, Dec. 2020, doi: 10.1109/TMI.2020.3005067.

C. Lombardini, A., Mytskaniuk, V., Sivankutty, S. et al. High-resolution multimodal flexible coherent Raman endoscope. *Light Sci Appl* 7, 10 (2018). <https://doi.org/10.1038/s41377-018-0003-3>

Reviewer #2 (Remarks to the Author):

This manuscript presents an interesting method to fabricate micro lenses for endoscopic OCT. By adjusting the wettability of the glass substrate, contact angle and radius of the microlens can be precisely adjusted. By adjusting the volume, diameter of the microlens can be adjusted. Sub-nanometer surface roughness can be achieved from both the curved and flat surfaces of the microlens, ensuring high optical quality of the imaging probe. Compared to previous methods using fiber ball lens and GRIN lens, the current process is controllable and repeatable, and avoids the tedious polishing step. It is suitable for batch manufacturing. The manuscript is well prepared with extensive details provided to reproduce the work. The topic is well within the scope of Communications Engineering. I am happy to recommend acceptance of the manuscript for publication after the authors properly address the

following comments.

1. Although designing the probe to work in the 800 nm wavelength range helps improve the image resolution and contrast, there is a tradeoff in imaging depth compared to 1300 nm. Please clearly describe this in the introduction and discussion.
2. For the demonstrated endoscopic probes, a wet angle of 90 degrees is used. It is not clear to me how to use microlenses with different contact angles. There might be some challenges of aligning the optical fiber to the microlens for non-90-degree contact angles, where input and output angle of the fiber may not be at the center of the flat surface. Please clarify.
3. Please add details on the procedures to align the fiber with the microlens. Would additional epoxy connecting the fiber and microlens affect the curvature and surface quality of the probe?
4. Since the probe is only ~1 mm in diameter, the rat esophagus shown in Fig. 5 may be collapsed on the imaging probe. Regions labeled as GEJ may be due to tissue folding and not the real gastroesophageal junction. The GEJ may be more clearly visible from the pullback direction and columnar structures, rather than layered structures, is expected to be seen from the gastric side. Please check.

Responses to the reviewers' comments

We are grateful to the reviewers' comments. We have revised the manuscript accordingly and provided the detailed responses as appended below. Author responses are in **blue**, changes and additions to the manuscript are in **red**, original texts in manuscript and reviewer's comments are in **black**, and all line numbers refer to the documents with redlined highlights.

Reviewer 1

General comments:

In this manuscript, the authors report the application of liquid-shaping method to the fabrication of sub-millimeter side-focusing ball lens that is well-known as essential to a miniature rigid or flexible OCT catheter. The authors explained their design logic and performance characterization and presented representative experimental results. The logic of the manuscript fits the scope of Comms. Eng., and the overall flow is easy to follow.

However, to minimize misleading information and to help future readers better appreciate the manuscript, I suggest the authors address the following comments.

Response:

We are grateful to the reviewer for his/her insightful and constructive comments. We have made careful revisions to our manuscript to address the concerns raised, with the aim of minimizing any potential misinterpretations and enhancing the overall clarity of our work. The specific changes made in response to each comment are detailed in the following sections.

Comment 1:

1. The initial and key word "self-assembled" of the title appeared only twice in the main text, which reflects on the lack of the evidence on why and how the microlens is self-assembled. The overall liquid-shaping-based fabrication procedure clearly necessitate extensive human intervention. Please reconsider the common meaning of "self-assembling", and then either provide more supportive explanation, or rectify the terminology to avoid misleading future readers.

Response:

Thank you for your insightful comment regarding the use of the term "self-assembled".

We initially used this term to highlight the inherent self-assembling behavior of the liquid lens on the substrate, which maintains stability and displays the desired surface profile due to the balance of surface tension and gravity, without the need for mechanical polishing or grinding.

We totally understand your concern about the potential for misunderstanding given the fine tuning of fabrication parameters in our fabrication process. Therefore, we have revised the title to "Liquid-shaped microlens for scalable production of ultrahigh-resolution OCT microendoscope". We believe this title more accurately reflects the contents of our manuscript and hope it will eliminate any potential confusion for future readers.

Original (see lines 1-2, page 1 in the manuscript):

"Self-assembled liquid-shaped microlens for scalable production of ultrahigh-resolution OCT microendoscope."

Revised (see lines 1-2, page 1 in the manuscript):

“Liquid-shaped microlens for scalable production of ultrahigh-resolution OCT microendoscope.”

Original (see lines 21-22, page 1 in the manuscript):

“This technique enables the flexible customization of self-assembled freeform microlenses...”

Revised (see lines 21-22, page 1 in the manuscript):

“This technique enables the flexible customization of freeform microlenses...”

Original (see lines 65-66, page 2 in the manuscript):

“...the rapid simultaneous fabrication of self-assembled side-focusing microlenses ...”

Revised (see lines 65-66, page 2 in the manuscript):

“...the rapid simultaneous fabrication of side-focusing microlenses ...”

Comment 2:

2. Since the authors compared their fabrication procedure to two-photon 3D microprinting, please provide more comparison information on pros and cons of these two methods, e.g., the cost of equipment (i.e., liquid-shaping v.s. two-photon microprinting), the cost of material (including the glass substrate with wetting property modified used in this manuscript), and complexity of freeform surface that could be achieved, so that future readers could better understand which technology to choose for their application.

Response:

We appreciate your constructive suggestion.

In response, we have further expanded our discussion to include a comparison of not only the liquid shaping-based and the two-photon 3D microprinting-based methods, but also the conventional GRIN fiber-based and the fiber ball-lens-based methods. This comprehensive comparison, which includes factors such as equipment cost, material cost, and complexity of the achievable freeform surface, is summarized in the updated Supplementary Table 1: Comparison of fabrication methods for OCT microendoscopes. We hope this additional information will assist future readers in selecting the most appropriate fabrication technology for their specific applications.

Original (see lines 194-195, page 15 in Supplementary Information):

“

Method	Fabrication time	Cost evaluation
GRIN fiber-based method	~ 3 hours, angle-polishing usually takes over 2 hours	Fibers, splicer, cleaver, and polishing tools needed, reduced yield rate by angle-polishing increases the ingredient and labor cost
Fiber ball-lens-based method	~ 3 hours, angle-polishing usually takes over 2 hours	Fibers, splicer, cleaver, and polishing tools needed, reduced yield rate by

		angle-polishing increases the ingredient and labor cost
Liquid shaping method	~ 90 minutes, no need for angle-polishing	Fibers, splicer, cleaver, and optical liquid needed, low cost

”

Revised (see lines 194-195, page 15 in Supplementary Information):

“

Methods	Pros	Cons
Liquid shaping-based method	 1. Enable monolithic design for minimization of OCT probe. 2. Freeform lens of high design freedom and rotational symmetric surface for aberration corrections. 3. Mass production of freeform lens. 4. Freeform lens of sub-nanometer surface roughness. 5. No need for lens polishing and precise optical alignment. 6. Scalable fabrication. 7. Short endoscope fabrication time of at most 1.5 hours. 8. Fabrication material cost of about \$70, including single-mode fiber, non-core fiber, optical liquid, glass substrate, wetting materials, toque coil, and protective sheath. 	 1. Need a relatively expensive liquid dispensing system of about \$30,000 or a home-made liquid dispensing system of about \$6,000.
Two-photon 3D microprinting-based method	 1. Enable monolithic design for minimization of OCT probe. 2. No need for lens polishing and precise optical alignment. 3. Freeform lens of high design freedom and complex asymmetric surface for aberration corrections. 	 1. Suboptimal surface roughness. 2. Lack of scalability potential. 3. Long printing time up to several hours, depending on the lens size. 4. Expensive two-photon printing machine, ownership cost of at least \$500,000. 5. Limited choice of photo resins.
GRIN fiber-based method	Enable monolithic design for minimization of OCT probe and GRIN fibers are commercially available.	 1. Limited choice of GRIN fibers and generally unknown fiber parameters. 2. Lack of capability to correct imaging aberrations. 3. GRIN fibers usually have strong chromatic and spherical aberrations at short wavelength regimes, such as 800 nm. 4. Requires costly and time-consuming angle-polishing. 5. Suboptimal surface roughness due to polishing. 6. Endoscope fabrication time up to several hours.
Fiber ball-lens-based method	 1. Enable monolithic design for minimization of OCT probe 2. Provide achromatic performance at short wavelength ranges, such as 800 nm. 	 1. Insufficient design freedom and controllability on ball-lens using fiber melting technique. 2. Requires costly and time-consuming angle-polishing.

		3. Suboptimal surface roughness due to polishing. 4. Limited to fabricate achromatic endoscopes of less than 1 mm in diameter. 5. Endoscope fabrication time up to several hours.
--	--	--

Comment 3:

3. The design rationale presented in Fig. 3 needs more clarification.

Comment 3.1:

3.1 The word “semi-spheroid” here is potentially misleading. A spheroid, also known as an ellipsoid of revolution or rotational ellipsoid, is a quadric surface obtained by rotating an ellipse about one of its principal axes; in other words, an ellipsoid with two equal semi-diameters. Also, the word spheroid originally meant "an approximately spherical body". Here, however, based on the description in line 166-167, what the authors seem to mean and used in their simulation is a semi-spherical lens “of radius r ...”. Please double check and clarify/revise accordingly.

Response:

We greatly appreciate your comment. You're correct in pointing out our misuse of the term "semi-spheroid" in the simulation section. Accordingly, we have revised this to "semi-spherical" throughout the manuscript to ensure our ideas are accurately conveyed. Thank you for helping us improve the clarity and precision of our work.

Original (see lines 168-169, page 7 in the manuscript):

“Zoomed-in view of the distal part of the microendoscope, consisting of an NCF of length L and a semi-spheroid microlens of radius r and utilizing an incident angle θ .”

Revised (see lines 168-169, page 7 in the manuscript):

“Zoomed-in view of the distal part of the microendoscope, consisting of an NCF of length L and a semi-spherical microlens of radius r and utilizing an incident angle θ .”

Original (see lines 189-191, page 8 in the manuscript):

“To demonstrate the feasibility of the liquid shaping technique for imaging performance optimization and aberration correction, we designed an 800-nm OCT microendoscope using a simple semi-spheroid microlens (Fig. 3a(iii)).”

Revised (see lines 189-191, page 8 in the manuscript):

“To demonstrate the feasibility of the liquid shaping technique for imaging performance optimization and aberration correction, we designed an 800-nm OCT microendoscope using a simple semi-spherical microlens (Fig. 3a(iii)).”

Original (see lines 240-242, page 10 in the manuscript):

“To illustrate the fabrication quality and controllability of the liquid shaping technique, the surface profile of the distal semi-spheroid microlens was first characterized using a 3D noncontact confocal surface profiler (MarSurf CM Expert, Mahr Inc.).”

Revised (see lines 242-244, page 10 in the manuscript):

“To illustrate the fabrication quality and controllability of the liquid shaping technique, the surface profile of the distal semi-spherical microlens was first characterized using a 3D noncontact confocal surface profiler (MarSurf CM Expert, Mahr Inc.).”

Original (see lines 496-499, page 18 in the manuscript):

“For the microendoscopes demonstrated in current work, the lens liquid with a volume of about 6.2 nL was dispensed on the substrate to fabricate the spheroid microlens of about 140- μm radius and contact angle of about 89.8°.”

Revised (see lines 495-498, page 18 in the manuscript):

“For the microendoscopes demonstrated in current work, the lens liquid with a volume of about 6.2 nL was dispensed on the substrate to fabricate the semi-spherical microlens of about 140- μm radius and contact angle of about 89.8°.”

Comment 3.2:

3.2 The source of astigmatism as presented in Fig. 3f is not clear. If the (total internal) reflection surface coincides with a great circle of the semi-spherical lens (i.e., the center of the spherical surface lies within the reflection surface), the entire optical train reduces to an on-axis geometry which should not suffer from astigmatism. If the reflection surface is offset from the great circle (i.e., from the centre of the sphere), which seems to be quite possible based on information presented in Fig. 1 and Fig. 2, this offset plays a more important role than NCF length and lens radius as presented in Fig. 3f, and therefore should be explicitly presented and Fig. 3f (and probably many other subfigures in Fig. 3) should be re-plotted with the main text revised accordingly also.

Response:

We appreciate the insightful comments from the reviewer.

In our raytracing simulation, we assumed the use of a "simple semi-spherical microlens" with a contact angle of 90°. We apologize for the confusion caused by the previous mention of a "simple semi-spheroid microlens". Thus, the scenario should align with the first case that the reviewer has pointed out, where "the reflection surface coincides with a great circle of the semi-spherical lens".

Further, we would like to clarify that the source of astigmatism is derived from the cylindrical wall of the glass capillary or plastic sheath, which has different curvatures parallel and perpendicular to the endoscope axis. This phenomenon has been recognized by the OCT community, as cited in references [19, 36-38].

We hope this explanation adequately addresses your concerns and we will ensure that our manuscript reflects this clarification.

Revised (see lines 207-209, page 8 in the manuscript):

“It is noted that the source of astigmatism is derived from the cylindrical wall of the glass capillary or plastic sheath, which has different curvatures parallel and perpendicular to the endoscope axis [19, 36-38].”

[19] Li, J. et al. Ultrathin monolithic 3D printed optical coherence tomography endoscopy for preclinical and clinical use. *Light Sci. Appl.* **9**, 124 (2020).

[36] Xi, J. et al. High-resolution OCT balloon imaging catheter with astigmatism correction. *Opt Lett* **34**, 1943-1945 (2009).

[37] Pahlevaninezhad, H. et al. Nano-optic endoscope for high-resolution optical coherence tomography in vivo. *Nat Photonics* **12**, 540-547 (2018).

[38] Woo Lee, M., Hoon Kim, Y., Xing, J. & Yoo, H. Astigmatism-corrected endoscopic imaging probe for optical coherence tomography using soft lithography. *Opt Lett* **45**, 4867-4870 (2020).

Comment 3.3:

3.3 Still about astigmatism, one important culprit is the equivalent cylindrical lens formed by the wall of the glass capillary tube or the plastic sheath. Whether such astigmatism source is accounted for is unclear, and little information is given regarding the refractive index or dimensions of the glass capillary (except an OD of 617 μm , line 523) or plastic sheath.

Response:

We are grateful for your constructive comment.

In accordance with the reviewer's comment, we concur that the astigmatism is indeed caused by the cylindrical lens effect of the glass capillary tube or plastic sheath. It is worth noting that this effect has been duly considered in our simulation, as depicted in Figure 3a (iii).

As the glass capillary is made of Silica and the material of plastic sheath is FEP, in our raytracing simulations we used the refractive indices of Silica and FEP in the material library of OpticStudio as below.

Refractive index profile of silica and FEP

Furthermore, we used the dimensions of glass capillary and plastic sheath in our raytracing simulations. In addition to the OD, in the revised manuscript we have included the wall thickness of both the glass capillary and plastic sheath.

Original (see lines 533-535, page 19 in the manuscript):

“Glass capillary of 617 μm in outer diameter was employed to protect the microendoscope during imaging.”

Revised (see lines 531-533, page 19 in the manuscript):

“Silica glass capillary (Molex LLC) of about 617 μm in outer diameter and about 40 μm in wall thickness was employed to protect the microendoscope during imaging.”

Original (see lines 538-539, page 20 in the manuscript):

“The flexible microendoscope was protected with a plastic sheath of a 626- μm outer diameter during imaging.”

Revised (see lines 536-538, page 20 in the manuscript):

“The flexible microendoscope was protected with a fluorinated ethylene propylene (FEP) plastic sheath (Zeus Inc.) of a 626- μm outer diameter and a 45- μm wall thickness during imaging.”

Comment 4:

4. Line 220, the word “radial angle” could be hard to understand. Please consider paraphrasing the description (e.g., how about azimuthal angle?)

Response:

Thanks for your suggestion and we agree that the term "azimuthal angle" would be more appropriate. We have updated the manuscript accordingly.

Original (see lines 219-221, page 9 in the manuscript):

“1D surface profiles of microlens extended by nonlinear least square fitting the measurements along four representative radial angles at 0°, 45°, 90°, and 135° (dashed lines in c).”

Revised (see lines 223-225, page 9 in the manuscript):

“1D surface profiles of microlens extended by nonlinear least square fitting the measurements along four representative azimuthal angles at 0°, 45°, 90°, and 135° (dashed lines in c).”

Original (see line 242, page 10 in the manuscript):

“The fitted one-dimensional surface profiles at four radial angles...”

Revised (see lines 246-247, page 10 in the manuscript):

“The fitted one-dimensional surface profiles at four azimuthal angles...”

Comment 5:

5. Does the ~2.4 μm axial resolution measured in air or in tissue? Please clarify.

Response:

Thanks for your question.

The axial resolution was measured in air. To clarify this, we have updated the manuscript.

Original (see lines 73-74, page 3 in the manuscript):

“...provided an ultrahigh resolution of $2.4 \mu\text{m} \times 4.5 \mu\text{m}$ (in axial and transverse directions, respectively).”

Revised (see lines 75-76, page 3 in the manuscript):

“...provided an ultrahigh resolution of $2.4 \mu\text{m} \times 4.5 \mu\text{m}$ (in axial and transverse directions in air).”

Comment 6:

6. Regarding chromatic aberration and astigmatism correction in side-viewing OCT probes, there have been previous efforts that should be cited; just to give a few examples:

A. Jiefeng Xi, Li Huo, Yicong Wu, Michael J. Cobb, Joo Ha Hwang, and Xingde Li, "High-resolution OCT balloon imaging catheter with astigmatism correction," *Opt. Lett.* **34**, 1943-1945 (2009)

B. Jessica Mavadia-Shukla, Payam Fathi, Wenxuan Liang, Shaoguang Wu, Cynthia Sears, and Xingde Li, "High-speed, ultrahigh-resolution distal scanning OCT endoscopy at 800 nm for in vivo imaging of colon tumorigenesis on murine models," *Biomed. Opt. Express* **9**, 3731-3739 (2018)

C. Li, K, Liang, W, Mavadia-Shukla, J, et al. Super-achromatic optical coherence tomography capsule for ultrahigh-resolution imaging of esophagus. *J. Biophotonics*. 2019; 12:e201800205.

Response:

Thank you. These suggested literatures have been added into our manuscript to provide a more comprehensive context.

The newly added reference:

[22] Mavadia-Shukla, J. et al. High-speed, ultrahigh-resolution distal scanning OCT endoscopy at 800 nm for in vivo imaging of colon tumorigenesis on murine models. *Biomed Opt Express* **9**, 3731-3739 (2018).

[23] Li, K. et al. Super-achromatic optical coherence tomography capsule for ultrahigh-resolution imaging of esophagus. *J Biophotonics* **12**, e201800205 (2019).

[36] Xi, J. et al. High-resolution OCT balloon imaging catheter with astigmatism correction. *Opt Lett* **34**, 1943-1945 (2009).

[37] Pahlevaninezhad, H. et al. Nano-optic endoscope for high-resolution optical coherence tomography in vivo. *Nat Photonics* **12**, 540-547 (2018).

[38] Woo Lee, M., Hoon Kim, Y., Xing, J. & Yoo, H. Astigmatism-corrected endoscopic imaging probe for optical coherence tomography using soft lithography. *Opt Lett* **45**, 4867-4870 (2020).

Comment 7:

7. It is worthy of noting that the liquid-shaped microlens fabrication protocol is applicable to fiberscopes based on other imaging modalities like confocal, two-photon, and coherent Raman, etc., that could also

benefit from advanced fiber-tip engineering. I suggest the authors include brief discussion about this topic to maximize the impact of their work, and here are some references to consider:

A. Jae-Beom Kim, Jaehun Jeon, Kyungmin Hwang, Daniel Y. Kim, Ki-Hun Jeong, "Objective-lens-free confocal endomicroscope using Lissajous scanning lensed-fiber," *J. Optical Microsystems* 1(3) 034501 (28 May 2021) <https://doi.org/10.1117/1.JOM.1.3.034501>

B. W. Liang et al., "Throughput-Speed Product Augmentation for Scanning Fiber-Optic Two-Photon Endomicroscopy," in *IEEE Transactions on Medical Imaging*, vol. 39, no. 12, pp. 3779-3787, Dec. 2020, doi: 10.1109/TMI.2020.3005067.

C. Lombardini, A., Mytskaniuk, V., Sivankutty, S. et al. High-resolution multimodal flexible coherent Raman endoscope. *Light Sci Appl* 7, 10 (2018). <https://doi.org/10.1038/s41377-018-0003-3>

Response:

We greatly appreciate your constructive suggestions.

We agree that the liquid shaping technique can be conveniently applied to other imaging modalities. We have added this point into the discussion part and cited relevant works.

Revised (see lines 441-443, page 17 in the manuscript):

“It is important to highlight that the proposed liquid shaping technique can be easily implemented in other types of fiberscopes that require a microlens on the fiber tip. This includes confocal, two-photon, and coherent Raman endoscopes, among others [52-54].”

Newly added references:

[52] Kim, J. B., Jeon, J., Hwang, K., Kim, D. Y. & Jeong, K. H. Objective-lens-free confocal endomicroscope using Lissajous scanning lensed-fiber. *J. Opt. Microsyst.* **1**, (2021).

[53] Liang, W. X. et al. Throughput-Speed Product Augmentation for Scanning Fiber-Optic Two-Photon Endomicroscopy. *IEEE Trans. Med. Imaging* **39**, 3779-3787 (2020).

[54] Lombardini, A. et al. High-resolution multimodal flexible coherent Raman endoscope. *Light-Sci Appl* **7**, (2018).

Reviewer 2

General comments:

This manuscript presents an interesting method to fabricate micro lenses for endoscopic OCT. By adjusting the wettability of the glass substrate, contact angle and radius of the microlens can be precisely adjusted. By adjusting the volume, diameter of the microlens can be adjusted. Sub-nanometer surface roughness can be achieved from both the curved and flat surfaces of the microlens, ensuring high optical quality of the imaging probe. Compared to previous methods using fiber ball lens and GRIN lens, the current process is controllable and repeatable, and avoids the tedious polishing step. It is suitable for batch manufacturing. The manuscript is well prepared with extensive details provided to reproduce the work. The topic is well within the scope of Communications Engineering. I am happy to recommend acceptance of the manuscript for publication after the authors properly address the following comments.

Response:

We appreciate the reviewer's comments and constructive feedback on our manuscript. Your recognition of the novelty and potential impact of our work is very encouraging. We have taken your comments into consideration and made appropriate revisions to our manuscript. Below, you will find our detailed responses to each of your points. Thank you for your valuable contribution to enhancing the quality of our work.

Comment 1:

1. Although designing the probe to work in the 800 nm wavelength range helps improve the image resolution and contrast, there is a tradeoff in imaging depth compared to 1300 nm. Please clearly describe this in the introduction and discussion.

Response:

We are grateful for your constructive comment. We acknowledge the importance of clearly describing the tradeoff between the imaging resolution and the imaging depth in OCT. In response to your comment, we have made the following revisions:

In the introduction,

Original (see lines 36-37, pages 1-2):

“This OCT offers a considerably high resolution (approximately 2 to 4 μm) and enhances image contrast.”

Revised (see lines 36-38, pages 1-2):

“This OCT offers a considerably high resolution (approximately 2 to 4 μm) and enhances image contrast, albeit at the expense of a shallower imaging depth compared to the 1300-nm OCT.”

In the discussion part, we have added,

Revised (see lines 371-373, page 15):

“In comparison to the 1300-nm OCT, the 800-nm OCT provides improved imaging resolution and contrast, albeit with a reduced imaging depth.”

Comment 2:

2. For the demonstrated endoscopic probes, a wet angle of 90 degrees is used. It is not clear to me how to use microlenses with different contact angles. There might be some challenges of aligning the optical fiber to the microlens for non-90-degree contact angles, where input and output angle of the fiber may not be at the center of the flat surface. Please clarify.

Response:

We appreciate your insightful query regarding the application of microlenses with different contact angles and the potential challenges in aligning the optical fiber to the microlens for non-90-degree contact angles.

One potential application could be to construct these microlenses directly on the flat surface of a fiber tip, which would result in a forward-view fiberscope (OCT or other imaging modalities, such as confocal) with controllable lens curvature, focal length, and depth of focus.

Regarding the alignment of the optical fiber and microlens with non-90-degree contact angles (suppose we still construct the side-view probe as discussed in the manuscript), we can still use the proposed high-precision four-dimensional stage with two inspection microscopes (providing top and side views, respectively). While we acknowledge that non-standard semi-spherical microlenses could present challenges to the alignment process, we are confident that these can be mitigated by using two microscopes with higher resolution and by implementing advanced profile analysis and fitting algorithms in the future. Thank you for bringing this important issue to our attention.

Comment 3:

3. Please add details on the procedures to align the fiber with the microlens. Would additional epoxy connecting the fiber and microlens affect the curvature and surface quality of the probe?

Response:

We appreciate the reviewer's suggestion and have now included additional details on the alignment procedures of the fiber and microlens in our manuscript.

In response to the question about the potential impact of the additional optical adhesive (NOA 81) on the curvature and surface quality of the probe, we would like to clarify that it does not have any adverse effects. The microlens, made of NOA81, is already polymerized prior to the alignment procedure, which means its shape is formed and fixed. The curved surface used for refraction along the light path is on the opposite side of the curved surface used for lens-fiber coupling, which is the area where the optical adhesive is applied to connect to the fiber tip. Hence, the curvature and quality of the curved surface that are critical for light refraction are not affected by the coupling optical adhesive.

Original (see lines 521-523, page 19):

“A high-precision four-dimensional stage with two inspection cameras was constructed and used to achieve accurate alignment between the fiber probe and microlens. About 0.5 nL of NOA 81 was applied on the fiber probe tip to optically bond the microlens. ”

Revised (see lines 517-523, page 19):

“To ensure precise alignment between the fiber probe and microlens, we developed a high-precision four-dimensional assembly stage equipped with two inspection microscopes that offer top and side views. This stage initially establishes a predetermined incident angle between the fiber probe and microlens. Subsequently, the fine adjustment of the fiber-lens incident angle and distance is achieved with the

assistance of the two microscopes. Following this, approximately 0.5 nL of NOA 81 is applied between the fiber tip and microlens to facilitate optical bonding, which is then cured using UV light.”

Comment 4:

4. Since the probe is only ~1 mm in diameter, the rat esophagus shown in Fig. 5 may be collapsed on the imaging probe. Regions labeled as GEJ may be due to tissue folding and not the real gastroesophageal junction. The GEJ may be more clearly visible from the pullback direction and columnar structures, rather than layered structures, is expected to be seen from the gastric side. Please check.

Response:

We appreciate the reviewer's expertise and valuable insight on this matter. Upon revisiting the 3D reconstructed visualization, we concur with the reviewer's observation that the imaged region is more likely to be folded esophageal tissues rather than the gastroesophageal junction (GEJ). To prevent any potential misunderstanding for future readers, we have removed the reference to the GEJ in the manuscript. Additionally, we have adjusted the flow of the original writing to ensure this section remains clear and easy to follow. Thank you for bringing this to our attention.

Original (see lines 274-279, page 11):

“

Fig. 5 Imaging rat esophagus using flexible microendoscope. a Cut-way view of a reconstructed 3D OCT image of a 36-mm-long rat esophagus adjacent to GEJ. **b** Representative 2D OCT image corresponding to the cross-section boxed with green dashed lines in **a**. **c** 3x close-up view of the region labeled with red dashed box in **b**. **d** Correlated hematoxylin and eosin (H&E) histology. GEJ: gastroesophageal junction, EP: stratified squamous epithelium, LP: lamina propria, MM: muscularis mucosae, SM: submucosa, CM: circular muscle, LM: longitudinal muscle. Scale bars are 250 μm .

”

Revised (see lines 277-281, page 11):

“

Fig. 5 Imaging rat esophagus using flexible microendoscope. a Cut-way view of a reconstructed 3D OCT image of a 36-mm-long rat esophagus. **b** Representative 2D OCT image corresponding to the cross-section boxed with green dashed lines in **a**. **c** 3x close-up view of the region labeled with red dashed box in **b**. **d** Correlated hematoxylin and eosin (H&E) histology. EP: stratified squamous epithelium, LP: lamina propria, MM: muscularis mucosae, SM: submucosa, CM: circular muscle, LM: longitudinal muscle. Scale bars are 250 μm .

”

Original (see lines 287, page 11):

“...and finally reached the section of the gastroesophageal junction (GEJ).”

Revised (see lines 289, page 11):

“...and finally reached the small esophagus.”

Original (see lines 291-293, page 11):

“The reconstructed 3D image revealed the GEJ section of the rat’s esophagus (Fig. 5a). A representative OCT cross-section clearly delineated the layered microstructure of the esophagus near the GEJ (Fig. 5b).”

Revised (see lines 293-294, page 11):

“The reconstructed 3D volumetric image and the representative OCT cross-section clearly revealed the layered tissue structures of rat’s esophagus (Fig. 5a and b).”

Original (see lines 297-302, pages 11-12):

“The GEJ regulates the flow of food and fluid between the stomach and esophagus and plays a key role in preventing acid reflux. Compared with conventional 1,300-nm OCT endoscopes, our 800-nm liquid-shaped OCT microendoscope facilitates ultrahigh-resolution imaging of the GEJ, potentially allowing the *in vivo* detection of early-stage Barrett’s esophagus, a condition mainly linked to gastroesophageal reflux disease.”

Revised (see lines 299-302, pages 11-12):

“Compared with conventional 1,300-nm OCT endoscopes, our liquid-shaped OCT microendoscope operating at 800 nm enables the acquisition of ultrahigh-resolution images of the fine microstructures in

the esophagus. This capability holds the potential for detecting subtle pathologies associated with early-stage diseases in vivo.”

REVIEWERS' COMMENTS:

Reviewer #1 (Remarks to the Author):

Thank the authors for handling all my comments with great care and detailed response. The manuscript is ready to publish.

Reviewer #2 (Remarks to the Author):

Good job with the revision.